# FLOW MATCHING FOR ONE-STEP SAMPLING

## ABSTRACT

Flow-based generative models have rapidly advanced as a method for mapping simple distributions to complex ones for which the distribution function is unknown. By leveraging continuous-time stochastic processes, these models offer a powerful framework for density estimation, i.e. an algorithm that samples new points based only on existing samples. However, their requirement of solving ordinary differential equations (ODEs) during sampling process incurs substantial computational costs, particularly for large amount of data and numerous time points. This paper proposes a novel solution, which is based on a theoretical analysis of Flow Matching (FM), to overcome this bottleneck, namely, we developed an algorithm to find the point prototype for a given point from the target distribution. By eliminating the need for ODE solvers, our method significantly accelerates sampling while preserving model performance. Numerical experiments validate the proposed approach, demonstrating its efficiency.

## 1 INTRODUCTION

The general idea of Continuous Normalizing Flow is to map one distribution to another by calculating a velocity field. By moving points from the source distribution along this velocity field, they converge to the target distribution. The Flow Matching (FM) approach (Lipman et al., 2023) enables the formulation of an efficient loss function to train a model representing the given velocity field.

Numerous approaches have been proposed for building models, defining loss functions, and implementing generative steps within FM. These include stochastic interpolants (Albergo & Vanden-Eijnden, 2023; Albergo et al., 2023), Rectified Flow (Liu et al., 2023), its accelerated variant (Liu et al., 2024), and Action Matching (Neklyudov et al., 2023). The FM approach has already been extended to various geometries (Chen & Lipman, 2024; Klein et al., 2023) and applications (Tamir et al., 2024; Jolicoeur-Martineau et al., 2023).

FM shares similarities with Diffusion Models (Sohl-Dickstein et al., 2015; Ho et al., 2020), which are at the forefront of generative deep learning tasks. However, a key difference lies in their modeling approach. While diffusion models utilize stochastic differential equations (SDEs) to compute the target distribution, FM models employ a deterministic approach, using ordinary differential equations (ODEs) to compute velocity fields that map the initial distribution to the target distribution. For generative tasks, a Gaussian distribution is commonly chosen as the initial distribution due to its well-understood mathematical properties and ease of sampling.

However, FM models have several training and sampling problems: These models require a long time to train due to the need to perform coupling for many pairs of sample points and long sampling time due to the need to solve the ODE during the inference procedure. For the training process, there is a lot of valuable work focused on better coupling algorithm (Tong et al., 2024a;b; Pooladian et al., 2023), using optimal transport (OT) mapping. We are eager to solve the problem of long sampling process to be able to generate huge amount of new data much faster. Approaches already exist to speed up the sampling process. Some of them again are connected with better coupling for less ODE solver steps (Wang, 2024), make faster sampling on pretrained models (Nguyen et al., 2024), adapting knowledge distillation (Salimans & Ho, 2022; Meng et al., 2022; Kim et al., 2024), search for the best stepsize (Li et al., 2023), better ODE solvers (Lu et al., 2022; Zheng et al., 2023).

In general, our idea is to make coupling pairs of points from the target (unknown) distribution with density $\rho_1$ which is represented as samples, and specially found points from the distribution with a given probability density $\rho_0$. The idea of such coupling comes from the application of the exact

formula for the vector field $v(x, t)$ (*i. e.* velocity of the point in the intermediate time), which is presented explicitly, in particular, in (Ryzhakov et al., 2024). The cited paper provides an explicit form of the vector field that minimizes the Flow Matching loss. By finding the trajectory of a point that starts from a given sample and moves with this velocity $v(x, t)$ taken with a minus sign, we can obtain a prototype[1] of the target sample. Overall, all exact prototypes are distributed according to the density $\rho_0$. Since the explicit expression for the velocity contains an integral over the target distribution, we cannot find the exact prototype, but we find it with a certain accuracy. However, as numerical experiments show, this accuracy is sufficient to train the model.

Our main contribution is the model training algorithm based on the exact expressions for the vector field in Flow Matching approach, in which training is performed on pairs of samples from the original and target distributions at once. These expressions allow us to make coupling of the source and target distribution points so that the resulting transformation is almost monotonic, *i. e.* the segments connecting possible pairs of samples almost never intersect. The models (neural networks), trained on these pairs, can generate new image very fast as these images are generated in one step of accessing the trained model. The proposed one-step approach can use coupling from any conditional mapping and is not limited to the chosen linear mapping. The disadvantage is a rather long process of finding these pairs at the training stage. Also, due to the fact that we only estimate the exact formula through samples and use invertible map with noise, the prototypes we obtain are not exactly the same as the exact prototypes; furthermore, the ODE solver that is used to find the prototypes has its own precision. However, these errors are moderated by the size of the buffer for evaluating the integrals in the exact formula for the velocity, and by tuning the parameters of the ODE solver.

## 2 PRELIMINARIES AND PROBLEM STATEMENT

We first briefly formulate the common task and known approaches to solve it.

### 2.1 CONTINUOUS NORMALIZING FLOWS

Consider two distribution with densities $\rho_0(x)$ and $\rho_1(x)$ of multivariate random variable $x \in \mathbb{R}^d$. Let $\psi_t(x_0)$ be a *flow* for $t \in [0, 1]$ that connect samples from the distributions $\rho_0$ and $\rho_1$. Consider time dependent *vector field* $v(x, t)$ such that

$$\begin{cases} \dfrac{\partial \psi_t(x_0)}{\partial t} = v\big(\psi_t(x_0), t\big), \quad \psi_0(x_0) = x_0, \end{cases}$$

and if $x_0 \in \mathbb{R}^d$ is a multivariate random variable having distribution $\rho_0$[2], the distribution of random variable $x_1 = \psi_1(x_0)$ must be approximately equal to the target distribution $\rho_1$.

Typically, initial distribution $\rho_0$ is given, and target distribution $\rho_1$ is unknown, and we have only access to samples from it. But there are also tasks where $\rho_0$ is unknown too, and we only have access to a (limited set of) samples from it.

For the given point, $x_0$ the flow $\psi_t$ defines a *trajectory* or a *path* $x(t) = \psi_t(x_0)$ with initial and final points $x_0$ and $x_1$, respectively.

A common approach is to approximate the vector field $v$ using a model (neural network) $v_\theta$, then sample a set $\{x_0\}$ of points from $\rho_0$ and solve a Cauchy problem for each $x_0$ from this set

$$\begin{cases} \dfrac{\mathrm{d}}{\mathrm{d}t}x(t) = v_\theta\big(x(t), t\big), \quad x(0) = x_0, \end{cases}$$

to obtain points $x_1 = x(1)$ that are being approximately distributed with $\rho_1$.

One of the approach for building $v_\theta$ is Conditional Flow Matching (Lipman et al., 2023; Tong et al., 2024a).

---

[1]Hereafter in the text we use the term "exact prototype" for those points of the original distribution that would be obtained by an absolutely accurate ODE solver given access to an absolutely accurate expression for the velocity. We use the term "prototype" to the approximated samples.

[2]Hereafter in the text we use the same notation for both the random distribution and its density function, unless this leads to ambiguities

## 2.2 Conditional Flow Matching (CFM)

We do not elaborate on the details of this approach and only note the main features that we need further.

The basic idea of the CFM approach is to use the so-called conditional map $\phi_{t,x_1}(x)$, which is a given function of time at two fixed endpoints: $\phi_{0,x_1}(x_0) = x_0$, $\phi_{1,x_1}(x_0) = x_1 + \epsilon(x_0)$ (the added small term $\epsilon$ depending on $x_0$ is sometimes needed for regularization so that the map is invertible). Based on this map, a conditional velocity (depending on the endpoint) is constructed as the time derivative of the map. And then, during training the model, random pairs of points are taken from the initial and target distributions, respectively, as well as randomly sampled time, and the model is trained at an intermediate point according to the selected map using the conditional velocity. A key advantage of the method is its theoretical proof of convergence to the desired target probability under specific conditions. The disadvantages of class CFM include large variance in the training loss and non-straightforward trajectories.

There are several ways to "straighten" trajectories, see cites in Introduction (Sec. 1) and Related Work (Sec. 5) sections. To reduce variance, several methods are also used, one of which is to use an explicit view in tractable form for a vector field (Ryzhakov et al., 2024). The cited paper proves that using this formula reduces variance under some conditions.

Our idea is to use this explicit form for $v$ to couple the samples.

# 3 Methodology

## 3.1 Main idea and Algorithm

**Explicit velocity** Our main idea is to find a prototype $X_0(x_1) \in \mathbb{R}^d$ of the given point $x_1 \in \mathbb{R}^d$ of the target distribution $\rho_1$ and then train a model for direct mapping from $X_0(x_1)$ to $x_1$. The operation of our algorithm is based on exact formulas for the velocity $v$, which we use in the form derived in (Ryzhakov et al., 2024). Namely, if we use invertible conditional map $\phi_{t,x_1}(x_0) = (1 - t)x_0 + tx_1 + \sigma t x_0$, the expression for the velocity is the following

$$v_\sigma(x, t) = \frac{\int \left(x_1 - x(1-\sigma)\right)\rho_0\left(\frac{x - x_1 t}{1 + \sigma t - t}\right)\rho_1(x_1)\,\mathrm{d}x_1}{(1 + \sigma t - t)\int \rho_0\left(\frac{x - x_1 t}{1 + \sigma t - t}\right)\rho_1(x_1)\,\mathrm{d}x_1}, \tag{1}$$

where $\rho_0$ is (unnormed) probability density function of the initial distribution and $\sigma$ is a small regularization parameter. In our experiments, we use the standard Gaussian distribution[3]:

$$\rho_0(x) \cong \exp(-\|x\|^2/2).$$

In the ideal case, when we know the distribution of $\rho_1$ or at least we can accurately take the integrals in the expression for the velocity, we would find its exact prototype $X_0(x_1)$ for each point $x_1$ from the target distribution of $\rho_1$. Then, by training a model $v_\theta$ (neural network or other model) on pairs $\{X_0(x_1), x_1\}$, we would immediately obtain a transformation from the initial distribution $\rho_0$ to the target smoothed distribution $\rho_1$[4].

**Importance Sampling** The formula (1) for the exact velocity contains integrals, where the integrand is multiplied by an unknown density $\rho_1$. In reality, we only have access to a certain set of samples from the $\rho_1$ distribution, so we can estimate these integrals with a given accuracy. Such a case is just suitable for the Importance Sampling method. Note that since we have to evaluate the integral standing in the denominators of the fraction, this evaluation may be biased (this is so-called self-normalized importance sampling, SIS). To get around this issue, one can use rejection sampling instead of SIS, as described in the above paper. Following the mentioned work (Ryzhakov et al., 2024), we estimate the integrals using importance sampling, since this approach gives good practical

---

[3]symbol $\cong$ means equality up to a constant factor

[4]as we use regularized map with $\sigma > 0$, then we actually get the distribution $\rho_1'(x) \cong \int \rho_0((x - y)/\sigma)\rho_1(y)\,\mathrm{d}y$ which at small $\sigma$ differs negligibly from the original distribution $\rho_1$ from a practical point of view, cf. Eq. (6) in (Lipman et al., 2023).

results even in the high dimensional case. Namely, in order to find a sample of point $x_1$, we take a sample set $\mathbb{B} = \{\overline{x}_1^k\}_{k=1}^N$ of size $N$, $\overline{x}_1^k \sim \rho_1$, that includes $x_1$, and use the following discretization of velocity $v_\sigma^{\text{dis}}$:

$$v_\sigma^{\text{dis}}[\mathbb{B}](x, t) = \sum_{k=1}^N \frac{\overline{x}_1^k - x(1-\sigma)}{1 - t + \sigma t}\big(\text{softmax}(Y^1, \dots, Y^N)\big)_k, \text{ where } Y^k = -\frac{1}{2}\frac{\left\|x - t \cdot \overline{x}_1^k\right\|_{L^2}^2}{1 - t + \sigma t}.$$

**Model training**    Using this velocity, we solve the following Cauchy problem

$$\left\{\frac{\mathrm{d}}{\mathrm{d}t}f(t) = v_\sigma^{\text{dis}}[\mathbb{B}]\big(f(t), t\big), \quad f(1) = x_1, \right. \tag{2}$$

for $t$ from 1 to 0 to find the prototype $X_0(x_1) = f(0)$ for a given $x_1$.

Such prototype-image pairs $\{X_0(x_1^l), x_1^l\}_{l=1}^n$ are constructed for a given batched size $n$ of samples of the target distribution $\rho_1$, with $n$ (significantly) smaller than $N$. Then we train the model $v_\theta$ using common quadratic loss

$$\text{loss} = \frac{1}{n}\sum_{l=1}^n \left\|v_\theta\big(X_0(x_1^l + \epsilon_l)\big) - x_1^l\right\|^2, \tag{3}$$

where i.i.d. variables $\{\epsilon_l\}$ are normally distributed with variance proportional to $\sigma$: $\epsilon_l \sim \mathcal{N}(0, \sigma \cdot \boldsymbol{I}_d)$.

We summarize this steps in Algorithm 1.

---

**Algorithm 1** One-step sampling training algorithm

---

**Require:** Sampler from distribution $\rho_1$ (or a set of samples); batch size $n$; size of buffer $N$ to estimate integrals; regularization parameter $\sigma$; model $v_\theta(\cdot)$; algorithm with parameters for stochastic gradient descent (SGD)
**Ensure:** quasi-optimal parameters $\theta$ for the trained model
1: Initialize $\theta$ (may be random)
2: Initialize buffer $\mathbb{B} \leftarrow \varnothing$ as empty set
3: **while** exit condition is not met **do**
4:     Sample set $\mathbb{X}$ of $n$ points $\mathbb{X} = \{x_1^i\}_{i=1}^n$ from target distribution $\rho_1$
5:     Add obtained points $\mathbb{X}$ to the buffer $\mathbb{B}$. If the size of the buffer exceeds $N$, remove the oldest points from it so that it contains $N$ points.
6:     Generate normal distributed noise $\epsilon \sim \mathcal{N}(0, \boldsymbol{I}_d)$
7:     For each point $x_1^i$ from $\mathbb{X}$ find the solution $X_0(x_1^i + \sigma \cdot \epsilon)$ of the Cauchy problem (2) with right-hand side $v^{\text{dis}}[\mathbb{B}]$ based on the points from the buffer $\mathbb{B}$.
8:     Update model parameters $\theta \leftarrow SGD(\theta, \text{loss})$ using loss in the form (3)
9: **end while**

---

At the inference step, we generate a point $x_0$ from the distribution $\rho_0$ and return the point $x_1 = v_\theta(x_0)$ immediately, without solving any differential equation.

## 3.2 EXTENSION: ADD LABELS

In case we have a dataset with labels, we can perform conditional generation. We use a conditional model $v_\theta(x_0, i)$ which receives as input, in addition to a point $x_0$ from the initial distribution, the label $i$ of a point which is an image of the given one.

When we solve the Cauchy problem (2), we use a different set of points for each of the labels for the buffer of $v^{\text{dis}}$. When calculating the loss, we also take into account the labels $\mathcal{L} = \{L_i\}_{i=1}^n$ of the sample points $\mathcal{X} = \{x_1^i\}_{i=1}^n$:

$$\text{loss} = \frac{1}{n}\sum_{l=1}^n \left\|v_\theta\big(X_0(x_1^l + \epsilon_l), L_l\big) - x_1^l\right\|^2. \tag{4}$$

We summarize this modifications in Algorithm 2.

---

**Algorithm 2** One-step sampling training algorithm with labels

---

**Require:** Sampler from distribution $\rho_1$ (or a set of samples); batch size $n$; size of buffer $N$ to estimate integrals; regularization parameter $\sigma$; model $v_\theta(\cdot, \cdot)$; number $m$ of labels; algorithm with parameters for stochastic gradient descent (SGD)

**Ensure:** quasi-optimal parameters $\theta$ for the trained model

1: Initialize $\theta$ (maybe random)
2: Initialize set of buffers $\{\mathbb{B}_i\}_{i=1}^m$ as empty sets: $\{\mathbb{B}_i \leftarrow \varnothing\}$ for $i = 1, 2, \ldots, m$
3: **while** exit condition is not met **do**
4:     Sample set of $n$ points $\mathbb{X} = \{x_1^i\}_{i=1}^n$ from target distribution $\rho_1$ with labels $\mathbb{L} = \{L_i\}_{i=1}^n$
5:     **for** $i = 1, 2, \ldots, m$ **do**
6:         Add points $\mathbb{X}[\mathbb{L} == i]$ from the whole set $\mathbb{X}$ with label $i$ to the buffer $\mathbb{B}_i$. If the size of the buffer $\mathbb{B}_i$ exceeds $N$, remove the oldest points from it so that it contains $N$ points
7:     **end for**
8:     Generate normal distributed noise $\epsilon \sim \mathcal{N}(0, \boldsymbol{I}_d)$
9:     For each point $x_1^i$ from $\mathbb{X}$ find the solution $X_0(x_1^i + \sigma \cdot \epsilon)$ of the Cauchy problem (2) with right-hand side $v^{\text{dis}}[\mathbb{B}_{L_i}]$ based on the points from the buffer $\mathbb{B}_{L_i}$ corresponding to this this point label
10:    Update model parameters $\theta \leftarrow SGD(\theta, \text{loss})$ using loss in the form (4)
11: **end while**

---

### 3.3 NEED TO USE $\sigma$

In our experiments, we took the value of $\sigma$ small ($\sim 10^{-2}$–$10^{-3}$) but not zero. The non-zero values of $\sigma$ makes the conditional map invertible. This is extremely important in our setup, as we solve the inverse ODE. In addition, we add a little noise to the original samples proportional to $\sigma$. Flow Matching approaches usually use a non-invertible map that corresponds to $\sigma = 0$. The intuition behind the use of non-zero $\sigma$ is that real-life datasets usually lie on a manifold of lower dimensionality than the dimensionality of the point space itself. Thus, it may also be that the prototypes lie on some low-dimensional manifold. But at the inference step, we feed arbitrary points to the model input. In this case, our model would not know how to behave at points where learning is fundamentally impossible. Thus, to artificially increase the dimensionality of the "prototype space", we add noise and use a regularized map.

Let us show the above issue on synthetic 2D examples, Fig. 1. In this example, the target distribution is a uniform distribution of two-dimensional points on the upper semicircle of a circle of radius 1.5. We generated $n = 200$ samples, for each sample we solved an ODE (2) with the right-hand side containing all the samples as set $\mathbb{B}$, thus $N = n$. To solve the ODE, we used the `solve_ivp` implementation of the Runge-Kutta method with an adaptive step that is controlled by the `tol` parameters from the `scipy` package. In all experiments, we added the same normally distributed noise $\epsilon$, which was multiplied by the $\sigma$ parameter.

One can see from Fig. 1 that when $\sigma$ is small and `tol` is insufficient, the point samples lie on a one-dimensional manifold. As `tol` decreases or $\sigma$ increases, the samples take the position characteristic of a normal distribution, as expected.

Our hypothesis is that for any value of $\sigma > 0$ it is possible to pick such `tol` value that the prototypes are distributed (for a sufficiently large number $N$) with the target distribution with a moderate accuracy; in contrast, for zero $\sigma$ at any `tol` this cannot be achieved. However, the authors do not have a rigorous proof of this statement yet.

## 4 EXPERIMENTS

### 4.1 TOY 2D EXAMPLES

We provide prove-of-concept experiments on toy 2D data, in particular for the `8 gaussians` dataset. During sampling, our method does not require solving ODE to transport points, it samples straightly from the model. Results for simple 2D cases are presented in Fig. 2.

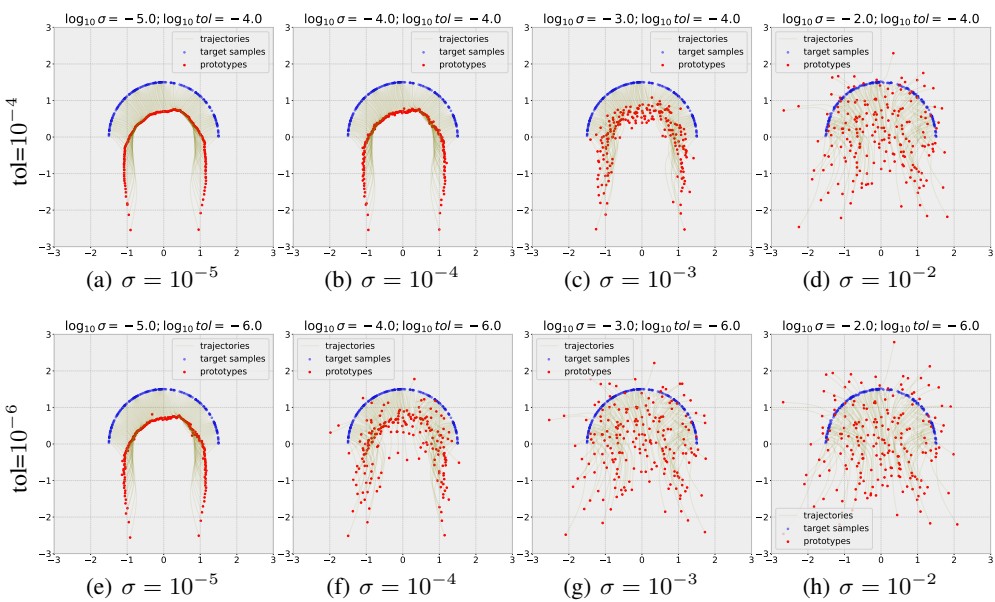

Figure 1: Prototypes for synthetic 2D data for different $\sigma$ values and different `tol` values of ODE solver, (a)–(d) $\sigma = 10^{-5}, 10^{-4}, 10^{-3}, 10^{-2}$ with `tol`$=10^{-4}$; (e)–(h) same, but `tol`$=10^{-6}$.

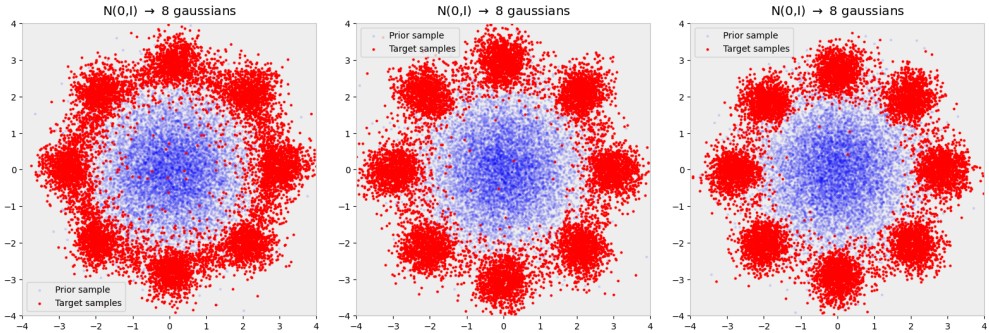

Figure 2: 2D `8 gaussians` examples results for 100, 500 and 1000 training steps.

## 4.2 IMAGE GENERATION

In Figure 3 we present images generated directly by sampling from Gaussian noise. For the prove-of-concept, we used a labeled **MNIST** dataset. For the training procedure we used Algorithm 2, for the model we used DiT (Peebles & Xie, 2022) due to the fact that for one-step sampling scheme we need more powerful neural network. We take $n = 128$, $N = 6 \cdot n$ and $m = 10$ in Algorithm 2, and Adam optimizer as SGD with lr$=10^{-3}$. Parameter $\sigma = 10^{-2}$. We take `odeint_adjoint` routine from `torchdiffeq` for solving Cauchy problem with `tol` $= 10^{-4}$. The number of training steps is $l = 15000$.

## 4.3 COLOR TRANSFER

For the color transfer problem, we consider the target distribution $\rho_1$ as a distribution of the given picture pixels considered as points in $\mathbb{R}^3$ space in the RGB model.

For the picture whose color we take as a basis, we train the model $v_\theta$ according to Algorithm 1. For the picture $P$ whose color we want to change, we also found pairs image-prototype according to Algorithm 1, but train the model $v_\chi$ to predict the prototype by the image. Thus, the loss for this

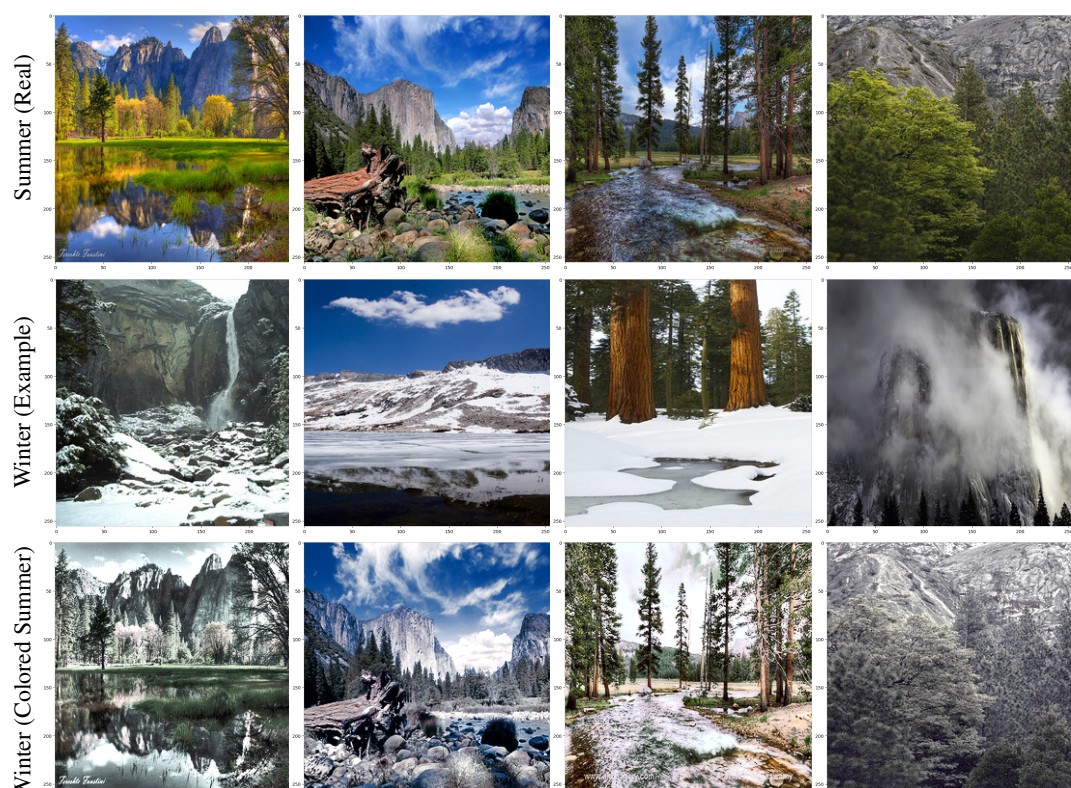

Figure 3: Result of **MNIST** dataset.

Summer (Real)

Winter (Example)

Winter (Colored Summer)

Figure 4: Colorization of the images of the **Winter2Summer** dataset. Up: initial image; middle: image with new color; down: colorized image

step is the following

$$\text{loss} = \frac{1}{n} \sum_{l=1}^{n} \left\| v_\chi(x_1^l) - X_0(x_1^l + \epsilon_l) \right\|^2 .$$

Note, number of pixels (number of samples in the target distribution) in the two pictures can be different.

On inference step, we simply took the composition of the two models $v_\theta(v_\chi(P))$ as the result picture.

We experiment on public **Winter2Summer** dataset (Zhu et al., 2017) containing 256x256 pixel images. The results are presented in Fig. 4.

**Implementation details**   We took Multilayer perceptron (MLP) as models $v_\theta$ and $v_\chi$. For $v_\chi$ we take MLP with two layers of 64 neurons each. For $v_\theta$ we take MLP with 1 layers of 64 neurons. We take $n = N = 128$ in Algorithm 1, and Adam optimizer as SGD with lr=$10^{-3}$. Parameter $\sigma = 10^{-2}$. We take `odeint_adjoint` routine from `torchdiffeq` for solving Cauchy problem with the default parameters. The number of training steps is $l = 500$. Note, that the total maximum number of samples $n \cdot l$ on which models are learned is less than the total number of pixels in each of the pictures ($256 \times 256$).

## 5  RELATED WORK

In this section, we only cite papers that discuss similar approaches. For details on Flow Matching theory, its modifications, connection of Flow Matching with Diffusion Models and other details on the subject we refer the reader to (Lipman et al., 2023; Tong et al., 2024a) and papers, cited in Introduction.

**Use of explicit formula**   To the best of our knowledge, the explicit formula for the velocity did not use for coupling points pairs before. In one form or another, the explicit form for the vector field has been mentioned, for example, in the following papers: (Liu et al., 2023; Neklyudov et al., 2023; Pooladian et al., 2023; Scarvelis et al., 2023; Xie et al., 2024).

**Coupling and trajectory straightening**   In the paper of Liu et al. (2023), the authors consider a way to accelerate the generation process, *i. e.*, the inference step, by iteratively training a new model based on the one obtained in the previous iteration. This approach leads to error accumulation, although a reduction in transportation cost has been proved for this approach. In addition, this paper mentions in the appendix the possibility of using an explicit formula (without regularization), only to accelerate the usual learning adopted in the Flow Matching framework, not to solve the inverse problem.

In (Kornilov et al., 2024) convex model (special type of neural network) and ideas based on the use of Shrödinger bridge are used to perform one-step generation of Flow Matching models. It turns out, that is it hard to learn such a model. In addition, the method presented in the cited paper has the same drawback as the original work on Conditional Flow Matching by Lipman et al. (2023), namely, the loss contains the expectation of both samples from $\rho_0$ and $\rho_1$ distributions, which, as shown in (Ryzhakov et al., 2024), leads to a large variance. Using an explicit formula for the vector field is one way around this obstacle.

Another approach of trajectory straightening was published in (Tong et al., 2024a). In this paper, a coupling based on minibatch Optimal Transport (OT-CFM) was proposed. However, this approach performs worse on large dimensions and, as shown in (Ryzhakov et al., 2024), is inferior in some examples to the simple use of an explicit formula (see Fig. 15 there). In addition, OT-CFM still solves ODEs at the inference step (although it is possible to solve ODEs on a coarser mesh due to more straighten trajectories), so this method reduces variance on the training step, but does not dramatically affect the generation step. Other OT-based approaches can be found in Pooladian et al. (2023) and in Related Work there.

## 6  CONCLUSION AND FUTURE WORK

The paper presents a method based on the solution of the Cauchy problem (2) in inverse time. As the right-hand side of the ODE, we consider the exact value of the velocity that minimize for Flow-Matching loss in the form from the paper Ryzhakov et al. (2024). Since we evaluate the integrals included in the formula for the exact velocity through Monte Carlo-like methods, namely, we use

importance sampling, the prototypes are not exact. However, the error in obtaining these prototypes is sufficient for the model (neural network) to be trained to predict the image immediately by the prototypes, bypassing the solutions of the differential equations.

We use a velocity expression (1) based on a reversible conditional map $\phi_{t,x_1}(x_0)$ with a regularization parameter $\sigma$. Using simple synthetic 2D examples, we show why regularization is necessary.

Our method can be easily extended to other conditional reversible maps, which can produced image-prototype pairing such that a neural network will learn better. The paper Ryzhakov et al. (2024) contains several examples of different exact formulas which can be incorporated in our Algorithm.

Also, one can use a model that assumes to be immediately gradient of convex transformation, as done in (Kornilov et al., 2024).

In addition to the formula with mapping from known distribution to unknown one, one can use the formula for the velocity in the case where both distributions are given only as samples. Explicit formulas in Sec. E.3.2 of (Ryzhakov et al., 2024) allows one to make such a coupling in this case too.

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
