# OpenReview forum: "Flow Matching for One-Step Sampling"
_ICLR.cc/2025/Conference — Submitted to ICLR 2025_

### Official Review · Reviewer_A9Ci · 2024-11-03

**Soundness:** 2
**Presentation:** 2
**Contribution:** 2
**Rating:** 3
**Confidence:** 3

**Summary:**

The paper presents a new flow matching approach for efficient sampling. The method finds an approximate prototype mapping by using importance sampling and the soft-max distribution. The prototype is finally found by solving a Cauchy problem. The prototypes and the connected target samples are used to define the  velocity field of the flow. The algorithm is evaluated on a 2D GMM task and a color transfer task.

**Strengths:**

- the new algorithm could potentially provide more computationally efficient generative models
- the presented algorithm seems to be reasonable, even though a few theoretical justifications are unclear

**Weaknesses:**

(1) there are no comparisons to other flow matching approaches or generative models (diffusion, consistency models) provided
(2) The evaluations are only done on a simplistic 2D dataset and the color transfer task, where I have difficulty assessing how good the result really is
(3) No evaluation and comparison of the computational efficiency of the generative model is offered (which was the main motivation of the approach). Its unclear to me how the single-step sampling here actually works? We still have the solve the ODE, don't we?
(4) Theory is also quite hand-wavy. For example, it is unclear to me how importance sampling is used to estimate the integral of Eq (1). Which distributions are replaced here ? I.e., what is the sampling distribution?
(5) It is unclear to me how tolerance and sigma interact and why we need sigma. From the presented theory, sigma would not be needed, would it? But without sigma, only very poor prototypes are learned, so this seems to be unsatisfactory to me

**Questions:**

- comparisons to other flow matching and consistency models (As they have the same goal) is needed
- Add more experiments for more complex target distributions
- Add more detail for the color tansfer task and compare to other methods (also qualitatitively)

See also weaknesses for more questions.

---

> ### Author Response · Authors · 2024-11-25
>
> **Thank you for the review. Here are our responses to your comments.**
>
> `(1) there are no comparisons to other flow matching approaches or generative models (diffusion, consistency models) provided (2) The evaluations are only done on a simplistic 2D dataset and the color transfer task, where I have difficulty assessing how good the result really is`
>
> Thank you for the comment. Our approach offers a significant advantage by generating results in a single step, without relying on the computationally expensive and complex processes of ODE or SDE solvers, nor does it involve distillation techniques. This sets it apart from existing methods that depend on these solutions.
>
> `(3) No evaluation and comparison of the computational efficiency of the generative model is offered (which was the main motivation of the approach). Its unclear to me how the single-step sampling here actually works? We still have the solve the ODE, don't we?`
>
> We have to solve ODE only in the training process to get couplings, in the sampling process we do not need to solve ODE. In the sampling step, we simply feed a random point that was obtained by sampling from the original Gaussian distribution $\rho_0$ to the input of the neural network and get a new image as output.
>
> `(4) Theory is also quite hand-wavy. For example, it is unclear to me how importance sampling is used to estimate the integral of Eq (1). Which distributions are replaced here ? I.e., what is the sampling distribution?`
>
> Thank you for your comment. We aimed to avoid overwhelming the reader with an overly theoretical tone, focusing on providing only the essential concepts to maintain clarity. Many of the details regarding importance sampling were already thoroughly discussed in the Ryzhakov et al. paper, where the exact formula is provided. However, we will revise the text to avoid any potential misunderstandings. Specifically, the formula (1) for the exact velocity involves integrals where the integrand is multiplied by an unknown density, $\rho_1$, the explicit form of which is not known but can be sampled from. This scenario is a typical application of importance sampling. It is important to note that since we must evaluate the integral in the denominator, this evaluation may be biased, leading to self-normalized importance sampling (SIS). To mitigate this issue, rejection sampling can be employed as an alternative to SIS, as outlined in the referenced article. We include this clarification in the revised version of the text.
>
> `(5) It is unclear to me how tolerance and sigma interact and why we need sigma. From the presented theory, sigma would not be needed, would it? But without sigma, only very poor prototypes are learned, so this seems to be unsatisfactory to me`
>
> The non-zero sigma makes the conditional map invertible. This is extremely important in our setup, as we solve the inverse ODE. If we immediately set sigma to zero, then even exact prototypes will not fill the entire $\rho_0$ distribution, since real data lie on a manifold whose dimensionality is less than the dimensionality of the space. In this case, our model will not know how to behave at points where learning is fundamentally impossible. Thus, we have to artificially add sigma and noise with the same variance so that the prototypes we get lie in the whole space.
> Note, however, that the smaller the sigma, the more accurate, theoretically, our solution becomes (the prototypes become closer to the exact prototypes). But with small sigma we have to take a smaller parameter ``tol'' to solve ODEs, which in turn affects the speed of the algorithm. Thus, here we have a tradeoff between accuracy and speed.
>
> **Let us answer your questions:**
>
> `comparisons to other flow matching and consistency models (As they have the same goal) is needed. Add more experiments for more complex target distributions.Add more detail for the color tansfer task and compare to other methods (also qualitatitively)`
>
> We would greatly appreciate any guidance on appropriate metrics for numerical comparison, particularly for the color transfer task, as, to the best of our knowledge, no existing method sufficiently measures the quality of color transfer.

---

> > ### Comment · Reviewer_A9Ci · 2024-11-27
> > **Reviewer response**
> >
> > Thanks for the rebuttal. The authors answered some of my concerns, however, the main criticism still remains (missing baselines, missing emperical comparison of computational complexity, more experiments with more complex target distributions are required). I will therefore stay with my rating.
> >
> > Regarding the question, I am not sure about a metric for th color transfer task as the task is newly introduced, but I guess any metric used for sampling can be used here as well (lower bound, effective sample size etc...). There might be good reasons why these metrics are not applicable, but this need to be discussed and at least some form of comparisons to other baseline methods need to be offered (even if thats only visually). Please refer to [1] for an overview of metrics and benchmark datasets.
> >
> > [1] Beyond ELBOs: A Large-Scale Evaluation of Variational Methods for Sampling
> > Denis Blessing, Xiaogang Jia, Johannes Esslinger, Francisco Vargas, Gerhard Neumann, Proceedings of the 41st International Conference on Machine Learning, 2024

---

> > > ### Author Response · Authors · 2024-12-03
> > >
> > > We thank the reviewer for the feedback. We believe that our method offers a promising approach to generative modeling, and we are committed to further improving its performance and addressing the limitations raised by the reviewer.
> > > As we use one-step generation, the complexity of the generation step coincide with one access to the model.
> > > While we appreciate the suggested metrics from [1], we believe that they may not be directly applicable to our task.

---

### Official Review · Reviewer_Xd3o · 2024-11-03

**Soundness:** 1
**Presentation:** 2
**Contribution:** 1
**Rating:** 1
**Confidence:** 4

**Summary:**

This paper proposes a one-step sampling method based on flow matching. Given a set of training samples from an unknown distribution $\rho_1$, a mapping is learned from a simpler known distribution $\rho_0$, typically a normal distribution, to $\rho_1$.  Using a discretized estimate of the flow, derived in (Ryzhakov et al. 2024), so-called prototypes in $\rho_0$ are found by solving an ODE for each training sample in $\rho_1$, under the assumption of linear flow. A network is then trained to directly map from $\rho_0$ to $\rho_1$, using the set of training samples paired with their respective prototypes.

**Strengths:**

The most important benefit of the proposed sampling approach is its speed, since instead of solving an ODE, sampling is done in one step using a neural network.

**Weaknesses:**

The experiments described in the paper are extremely limited. It is shown that 8 Gaussians can be generated using the proposed sampling method. It is also shown, with four example images, how colors can be transferred from one distribution to another, with target colors given by a separate image. Furthermore, in the paper, the sampling method was not compared to any other alternative method or ablated versions of the same method. For a publication to be recommended, the experiments should be expanded to include comparisons against relevant baselines. The text describing the color transfer experiments should preferably also be rewritten since the current version is too unclear.

It is rather unclear what a prototype is, whether it is the point in $\rho_0$ one would converge to if there are no errors in ODE solver, or whether a prototype can be any point in $\rho_1$ that you happen to converge to. Something that would have been interesting to know is how precise the prototypes truly are, and what effect the errors have on the end result. It is claimed that the quality of the prototypes is sufficient for images to be predicted, but this is a question that ought to be studied in greater depth.

When prototypes are found in $\rho_0$, noise is first added to the position in $\rho_1$, but the motivation for this is hardly satisfactory. In almost a full page the paper tries to argue why the added noise is necessary. It seems that the implementation of the Runga-Kutta method relies on a particular set tolerance parameter. If the tolerance is set too high, the method will stop early, and points will never reach $\rho_0$. However, if normal distributed noise is added in $\rho_1$, at least the points end up being spread like a normal distribution, but there is no convincing argument that the errors would then be smaller.

The language of the paper is unfortunately rather problematic with numerous missing articles (the, a), improper prepositions, and awkward sentences that are hard to interpret. However, the problems are not too severe for the material to be understood and should be relatively easy to correct in a final version of the paper.

The two algorithms are very similar to each other. The only difference seems to be that if you have a discrete label, a separate mapping is learned for each value of the label. It would have been sufficient to just keep Algorithm 2.

The notation for buffer B varies in different parts of the text.

**Questions:**

* What is a prototype? How would a prototype be defined? For each point in $\rho_1$ are there many potential prototypes?
* Does the $\sigma$ in (1) have anything to do with the $\sigma$ in (3)? Isn’t the $\sigma$ in (1) necessary to make the mapping invertible, while the $\sigma$ in (3) is related to the tolerance?
* Why is no noise added in (4), just like in (3)?

**Details Of Ethics Concerns:**

There are no ethical issues.

---

> ### Author Response · Authors · 2024-11-25
>
> **We value your feedback. We will now address your comments in detail.**
>
> `The experiments described in the paper are extremely limited. It is shown that 8 Gaussians can be generated using the proposed sampling method...`
>
> Direct quantitative comparisons are difficult to make because there are no standard metrics for evaluating 2D data or MNIST dataset. Nevertheless, we argue that a key strength of our approach lies in its ability to generate results in a single step, without the need for an ODE or SDE solver. This sets our method apart from many others, which depends on such solvers. Moreover, our method does not rely on distillation techniques. We would appreciate any suggestions for appropriate metrics to assess performance on these particular datasets. We have added MNIST experiments, rewritten the color transfer experiments and included in the new PDF file.
>
> `It is rather unclear what a prototype is, whether it is the point in $\rho_0$ one would converge to if there are no errors in ODE solver, or whether a prototype can be any point in $\rho_1$ that you happen to converge to...`
>
> We thank the reviewer for the comment. We would be grateful for more specific feedback from the reviewer regarding any aspects of the prototype generation process that may be unclear or require further elaboration.
>
> We have added a footnote to the Introduction to further clarify this point. Additionally, we have expanded our answer about prototypes in the subsequent discussion below. We'll make a deeper investigation into the properties of prototypes in a future research.
>
> `When prototypes are found in $\rho_0$, noise is first added to the position in $\rho_1$, but the motivation for this is hardly satisfactory...`
>
> The non-zero sigma makes the conditional map invertible. This is extremely important in our setup, as we solve the inverse ODE. If we immediately set sigma to zero, then even exact prototypes will not fill the entire $\rho_0$ distribution, since real data lie on a manifold whose dimensionality is less than the dimensionality of the space. In this case, our model will not know how to behave at points where learning is fundamentally impossible. Thus, we have to artificially add sigma and noise with the same variance so that the prototypes we get lie in the whole space.
> Note, however, that the smaller the sigma, the more accurate, theoretically, our solution becomes (the prototypes become closer to the exact prototypes). But with small sigma we have to take a smaller parameter ``tol'' to solve ODEs, which in turn affects the speed of the algorithm. Thus, here we have a tradeoff between accuracy and speed.
>
> `The language of the paper is unfortunately rather problematic with numerous missing articles (the, a), improper prepositions, and awkward sentences that are hard to interpret. However, the problems are not too severe for the material to be understood and should be relatively easy to correct in a final version of the paper.`
>
> Thank you for the comment, we carefully read the text and corrected found errors, rewrote some parts for better understanding.
>
> `The two algorithms are very similar to each other. The only difference seems to be that if you have a discrete label, a separate mapping is learned for each value of the label. It would have been sufficient to just keep Algorithm 2.`
>
> To maintain clarity and simplicity, we have presented the first algorithm in its core form. It directly highlights the fundamental aspects of our method and can be readily applied in practical settings.
>
> `The notation for buffer B varies in different parts of the text.`
>
> We thank the reviewer for pointing out the notational error. We have corrected this in the revised paper.
>
> **Let us answer your questions:**
>
> `What is a prototype? How would a prototype be defined? For each point in $\rho_1$ are there many potential prototypes?`
>
> Thank you for the question. Mathematically, we define prototype as $X_0(x_1)\in\mathcal{R}^d$, which in practice is just a point that we get after solving the ODE.
> Yes, each point may have many prototypes and the prototype of the given point depends on the tolerance of the ODE solver.
> We use the phrase ``exact prototype'' when we mean the point in $\rho_0$ we get to if there are no errors in ODE solver and if we know $\rho_1$ exactly.
>
> `Does the $\sigma$ in (1) have anything to do with the $\sigma$ in (3)? Isn’t the $\sigma$ in (1) necessary to make the mapping invertible, while the $\sigma$ in (3) is related to the tolerance?`
>
> If we use invertible map, then under conditional map the initial Gaussian $\rho_0$ will move to the Gaussian distribution located near the point $x_1$ with variance $\sigma$. To model the inverse situation, we add a noise with the same variance to the points from the target distribution. Thus, $\sigma$ in (1) and the variance of $\epsilon_l$ in (3) are the same.
>
> `Why is no noise added in (4), just like in (3)?`
>
> Thank you for your comment, it was a typo, we modified the text.

---

> > ### Comment · Reviewer_Xd3o · 2024-11-27
> >
> > This reviewer will stick to the same recommendation. Despite the addition of the MNIST experiment the experimental secrion is very limited, with no comparisons to alternative methods. Furthermore, the link between the $\sigma$ in (1) and (3) is still very unclear. Regardless of motivation given, the noise term depends on $x_0$ in (1), but not in (3).

---

> > > ### Author Response · Authors · 2024-12-03
> > >
> > > Thank you for the feedback. In (1) we have no noise term as this is the exact formula based on distributions. But if we consider invertible map and Gaussian noise as initial distribution, the distribution at $t=1$ will be not the target distribution, but a smooth approximation to it.
> > > This is equivalent to saying that we have added a small Gaussian noise (with small variance $\sigma$) to the target distribution. To simulate this situation, we add such noise with small sigma to $x_1$.
> > > We believe that the mathematical explanation provided in the paper is sufficient.
> > > However, we will consider adding a more explicit derivation or discussion of this transition in the Appendix.

---

### Official Review · Reviewer_7Roq · 2024-11-04

**Soundness:** 2
**Presentation:** 1
**Contribution:** 1
**Rating:** 3
**Confidence:** 2

**Summary:**

This paper presents a flow-based generative modelling approach that does not require solving an ODE for inference. The method seems to rely heavily on the "exact velocity" from (Ryzhakov et al., 2024); however, both the motivation and details of this need more development. Moreover, more comparison (especially numerical results) to previous works is needed. E.g. other approaches that bypass inference integration, like consistancy models, are not mentioned in the related work.

**Strengths:**

1. Adresses the problem of long inference time by bypassing the inference ODE integration.

**Weaknesses:**

1. More elaboration on the method of Ryzhakov would be helpful.
2. Unclear about motivation and comparison to related works.
3. Lack of numerical results.

**Questions:**

1. Where exactly is the convexity in the problem?
2. How does this method compare to related work?

---

> ### Author Response · Authors · 2024-11-25
>
> **We appreciate your review. Let's discuss your specific comments.**
>
> `More elaboration on the method of Ryzhakov would be helpful.`
>
> Thank you for the comment. We provide a concise overview of the main ideas of the Ryzhakov method in the introduction to maintain the paper's readability. A more detailed mathematical exposition is presented in Section 3.
>
> `Unclear about motivation and comparison to related works.`
>
> Our motivation is that one step allows one to  much faster  generate images, without numerically solving ODE or PDE. We write this in the introduction section. `We are eager to solve the problem of long sampling process to be able to generate huge amount of new data much faster.` We expanded the “Introduction” section to focus more on the problem statement and limitation of our method.
>
> `Lack of numerical results.`
>
> We would appreciate any suggestions from the Reviewer regarding suitable quality metrics for 2D cases and image generation. However, we believe that the ability to perform one-step sampling without relying on heuristic techniques such as distillation is a significant advantage of our method, as demonstrated by the promising MNIST results included in the updated PDF.
>
> **Let us answer your questions:**
>
> `Where exactly is the convexity in the problem?`
>
> Since one-step generation in our case is analogous to straight trajectories in classical Flow Matching when solving ODEs, we expect that these trajectories would not overlap for all possible model inputs. This could potentially imply the monotonicity of our model, suggesting that it might be the gradient of a convex function. However, as we have not delved deeper into this aspect, we have removed the mention of convexity from the text. Thank you for your insightful comment.
>
> `How does this method compare to related work?`
>
> While direct quantitative comparisons are challenging due to the absence of established metrics for 2D data and MNIST datasets, we believe our method's ability to perform one-step generation without an ODE solver is a significant advantage. We would be grateful to receive any feedback regarding suitable metrics for evaluating these specific data.

---

> > ### Comment · Reviewer_7Roq · 2024-12-03
> > **Thanks for your reply**
> >
> > I thank the authors for their response. I will keep my score primarily because I did not feel there was sufficient comparison (in related work and results) to other models with the same motivation of bypassing integration steps. This information would be helpful to contextualize the value of this model. Lastly, I still feel that this paper assumes that the reader already knows about the cited Ryzhakov paper, which seems unlikely at this time. Overall, I also concur with the points that the other reviewers raised.

---

> > > ### Author Response · Authors · 2024-12-03
> > >
> > > We thank the reviewer for the comment. We believe that the provided explanations of Ryzhakov et al, along with the references to the relevant literature, are sufficient for understanding the core concepts of the method, we will consider adding any particular explanations to the Appendix.

---

### Official Review · Reviewer_pKwW · 2024-11-04

**Soundness:** 2
**Presentation:** 1
**Contribution:** 3
**Rating:** 6
**Confidence:** 2

**Summary:**

This paper considers continuous normalizing flows and proposes an algorithm for finding the corresponding point in the base distribution (called its prototype) for a given point from the target distribution, without having to solve the defining ODE. For each sampled target point, the explicit velocity function is followed backwards to find a suitable base distribution point. Using the set of point pairs as a training set, a model is trained on them; that is, the generation can take place in a single step, and the network does not need to be invertible.

**Strengths:**

The single step sampling results in a greatly improved speed without significantly affecting the model performance.

**Weaknesses:**

Language: there are many spelling and grammatical errors, and the sentences are sometimes incomprehensible.
Especially in the introduction, the main ideas are hard to comprehend, especially if one does not know the previous work in detail.
The experiment in Section 4.2 should be explained in more detail or a reference needs to be added if it is a common method for image colorization.
Line 114: there is no "Introduction and Related Work" section, these are two separate sections in the paper.

**Questions:**

1. Sec 3.3: what does the >>tol<< parameter have to do with the generation? Why and how does it affect the shape of the base distribution?
2. Page 3, footnote 2: "up to a constant": Is it a constant factor or a constant term?

---

> ### Author Response · Authors · 2024-11-25
>
> **Thank you for your review. We will now respond to your comments.**
>
> `Language: there are many spelling and grammatical errors, and the sentences are sometimes incomprehensible. Especially in the introduction, the main ideas are hard to comprehend, especially if one does not know the previous work in detail. The experiment in Section 4.2 should be explained in more detail or a reference needs to be added if it is a common method for image colorization. Line 114: there is no "Introduction and Related Work" section, these are two separate sections in the paper.`
>
> We thank the reviewer for the comments. We have carefully revised the paper to address the identified issues, including correcting grammatical errors and rewriting line 114 for improved clarity. We have also tried to provide a comprehensive overview of the necessary background information in the preliminaries and introduction sections. To maintain readability, we have avoided overly detailed mathematical explanations and paper overviews, as these are already covered in the cited works. We would be grateful for specific feedback on any particular areas that require further elaboration.
>
> We have improved the Experiments section, particularly regarding the color transfer task. To our knowledge, there are currently no established metrics or common approaches for evaluating color transfer performance. We believe that our method's ability to perform one-step sampling without relying on distillation, a heuristic technique, is a significant advantage. This is further supported by the promising MNIST results included in the updated PDF.
>
> **Let us answer your questions:**
>
> `Sec 3.3: what does the >>tol<< parameter have to do with the generation? Why and how does it affect the shape of the base distribution?`
>
> We thank the reviewer for the question. The tolerance parameter (tol) in our work controls the accuracy of the numerical ODE solver employed to find pairs of points from different distributions. In essence, the ODE solutions serve as prototypes for target points. Figure 1 visualizes the target points and their corresponding prototypes obtained through the ODE solver, rather than the generated points themselves. Poor ODE solver parameters (tol, $\sigma$) lead to inaccurate prototypes, which in turn result in poor training data and ultimately poor generation quality.
> In the inference step, i.e., image generation, tol is not used since we are not solving ODEs.
>
> `Page 3, footnote 2: "up to a constant": Is it a constant factor or a constant term?`
>
> Up to a constant factor, i.e., not normalized. We added the clarification to the text.

---

> > ### Comment · Reviewer_pKwW · 2024-11-27
> >
> > I thank the authors for their response.
> >
> > **On the question of color transfer**
> >
> > I think the problem description is still missing. (The Section starts with "For *the* color transfer problem" (emphasis mine), which would indicate that it has been already introduced.)
> >
> > **On the question of the parameter `tol`**
> >
> > So this is basically a hyperparameter that needs to be tuned. This should be explicitly written in the paper.
> > I think at least the claim `tol` affects ODE solver accuracy" should be mentioned.

---

> > > ### Author Response · Authors · 2024-12-03
> > >
> > > We thank the reviewer for the answer.
> > > We will add the clarification to the text, that this is the parameter to be tuned.
> > > But we want to note, that the tolerance is a standard parameter used in various numerical methods, including ODE solvers, and its general function is well-understood in the field.

---

### Author Response · Authors · 2024-11-25

We thank the reviewers for their valuable feedback. We have revised the paper to address the identified issues, including correcting grammatical errors, improving notation, and adding MNIST results to illustrate the effectiveness of our method. We have also incorporated additional explanations to clarify certain aspects of the work. We have uploaded an updated pdf file.

We believe that the ability to perform one-step sampling without distillation and without the need of solving ODE or SDE is a significant advantage of our approach, as demonstrated by the promising MNIST results included in the updated PDF.

A comprehensive numerical  and theoretical analysis is a subject of the future work.

---

### Meta-Review · Area_Chair_LUCe · 2024-12-20

**Metareview:**

This work proposes a sampling technique for Flow Matching, a recently emerging approach for generative modeling. Its main novelty is a sampling algorithm that, unlike in previous work, does not require solving an ordinary differential equation. It is based on a model training algorithm that uses exact expressions for the vector field in flow matching and training on samples from the original and target distributions simultaneously. One weakness of this work is its lack of clarity. For example, while some of the terminology may be standard in a subfield, the work may have also needed more background to be broadly accessible to the NeurIPS community.  Another concern raised by the reviewers is the lack of comparisons and a very limited experimental section. While the first weakness was slightly mitigated in the rebuttal process, the work still remains somewhat difficult to access, and the reviewers remained concerned about experimentation and comparisons, which is why I recommend rejection.

**Additional Comments On Reviewer Discussion:**

The authors made a good effort to address most of the reviewer's concerns on writing, while the concerns about comparisons and experimentation were insufficiently addressed, as stated by the reviewers in their concluding comments. Ultimately, it feels that the authors are right that it is a big advantage to avoid the ODE computation. As this advantage is the main claim of the work, substantiating that claim by additional experiments is a reasonable expectation on the reviewers' side.

---

### Decision · Program_Chairs · 2025-01-22

Reject